# Rapid, high-throughput phenotypic profiling of endosymbiotic dinoflagellates (Symbiodiniaceae) using benchtop flow cytometry

**Colin Jeffrey Anthony**⍟*, **Colin Lock**⍟, **Bastian Bentlage**⍟*

Marine Laboratory, University of Guam, Mangilao, Guam, United States of America

* colin_anthonynw@outlook.com (CJA); bentlageb@triton.uog.edu (BB)

## Abstract

Endosymbiotic dinoflagellates (Family Symbiodiniaceae) are the primary producer of energy for many cnidarians, including corals. The intricate coral-dinoflagellate symbiotic relationship is becoming increasingly important under climate change, as its breakdown leads to mass coral bleaching and often mortality. Despite methodological progress, assessing the phenotypic traits of Symbiodiniaceae *in-hospite* remains a complex task. Bio-optics, biochemistry, or "-omics" techniques are expensive, often inaccessible to investigators, or lack the resolution required to understand single-cell phenotypic states within endosymbiotic dinoflagellate assemblages. To help address this issue, we developed a protocol that collects information on cell autofluorescence, shape, and size to simultaneously generate phenotypic profiles for thousands of Symbiodiniaceae cells, thus revealing phenotypic variance of the Symbiodiniaceae assemblage to the resolution of single cells. As flow cytometry is adopted as a robust and efficient method for cell counting, integration of our protocol into existing workflows allows researchers to acquire a new level of resolution for studies examining the acclimation and adaptation strategies of Symbiodiniaceae assemblages.

## 1. Introduction

Symbiodiniaceae are dinoflagellates known for their endosymbiotic relationship with many marine invertebrates, most notably reef building corals [1]. The breakdown of this symbiosis can lead to coral bleaching and mortality [2]. Despite methodological progress identifying functional and genetic variation in Symbiodiniaceae, assessing trait variation of Symbiodiniaceae *in-hospite* remains a complex task [3]. Pulse Amplitude Modulated (PAM) fluorometry is widely used to quantify photosynthetic efficiency of Symbiodiniaceae photosystems *in-hospite* [4] both *in situ* [5] and *ex situ* [6]. However, PAM fluorometry provides an aggregate measure of photosystem performance for the Symbiodiniaceae population and does not quantify variation in photosystem performance between cells within the symbiont population. Isolation, characterization, and quantification of photopigments using spectrometry or fluorometry

**Data Availability Statement:** All relevant code and data were deposited on GitHub and archived on Zenodo (https://doi.org/10.5281/zenodo.8260110).

**Funding:** This study was supported by National Science Foundation (https://www.nsf.gov/) award OIA-1946352. The funders had no role in study design, data collection and analysis, decision to publish, or preparation of the manuscript. Any opinions, findings, conclusions, or recommendations expressed in this material are those of the author(s) and do not necessarily reflect the views of the National Science Foundation.

**Competing interests:** The authors have declared that no competing interests exist.

struggles with similar resolution issues, relying on the extraction of pigments from cells prior to analysis, followed by the normalization of measured pigment quantities to observed cell densities or overall protein content [7]. Microscopic methodologies are capable of quantifying phenotypic differences of single Symbiodiniaceae cells, but these methodologies are labor intensive, able to process only a small number of cells, and often require multiple preparation methods or instruments to collect different phenotypic traits [8, 9]. By contrast, flow cytometry is a rapid method to quantify phenotypic variation within a Symbiodiniaceae population [10–12] by generating thousands of phenotypic profiles simultaneously and requiring comparatively little sample preparation time. Phenotypic profiles generated by flow cytometry include metrics associated with cell shape, cell size, and photopigments. Flow cytometry is commonly used to count Symbiodiniaceae cells [13], but the additional phenotypic data recorded by the instrument has thus far been largely ignored in Symbiodiniaceae research. To facilitate the use of these data, we provide a detailed protocol and conducted sensitivity analyses to provide guidance for sample preparation for flow cytometric phenotypic profiling of Symbiodiniaceae.

Flow cytometry was developed in 1968 [14] and has experienced massive advances in the last 50 years moving from a method for cell counting to detecting fluorescent protein markers, detecting RNA expression, describing cell cycles by staining DNA, studying signal pathways with antibodies, sorting cells, and quantifying protein content with fluorescent tags [15]. In the early 1980s, several groups began to highlight the power of flow cytometry for phytoplankton research, demonstrating its ability to characterize phytoplankton communities' taxonomic composition using photopigment autofluorescence, cell size, and cell shape [16, 17]. Flow cytometry has also been employed for the rapid quantification of autofluorescent photopigments by measuring the relative strength of their fluorescent signals [10, 11, 17–23]. Within coral biology, flow cytometry is widely used for determining Symbiodiniaceae cell density [13], but has also been used to identify functional groups and phenotypes of Symbiodiniaceae [10, 24], quantify autofluorescent pigments [11, 18, 19], enumerate endosymbiotic bacterial communities [25, 26], and even examine cell cycle dynamics [27–29]. Built from these concepts and methodologies, we developed a straight-forward flow cytometry protocol that generates consistent phenotypic profiles for thousands of individual Symbiodiniaceae cells to quantify the phenotypic variation of the entire assemblage, a currently difficult to access level of resolution. Sample degradation may influence the metrics collected by our protocol and we provide recommendations for sample preparation and handling based on sensitivity analyses conducted by us. Our protocol has the potential to reveal phenotypic differences between Symbiodiniaceae communities and may help elucidate whether phenotypic variation in Symbiodiniaceae assemblages is caused by phenotypic plasticity of Symbiodiniaceae cells or changes in community composition (e.g. shuffling to new functionally advantageous lineages). With proper implementation and considerations of methodological limitations, the protocol described here can be used to expediently identify functional differences in Symbiodiniaceae assemblages between host species, sampling sites, timepoints, and potentially phenotypic variation and niche partitioning of Symbiodiniaceae communities within a single host.

## 2. Materials and methods

### 2.1 Protocol

The protocol described in this article identifies Symbiodiniaceae cells based on their high red autofluorescence (Fig 1A), and then uses forward scatter, side scatter, red fluorescence, and green fluorescence to generate phenotypic profiles for individual Symbiodiniaceae cells from an endosymbiotic assemblage. The protocol was developed using the Guava easyCyte 6HT-2L

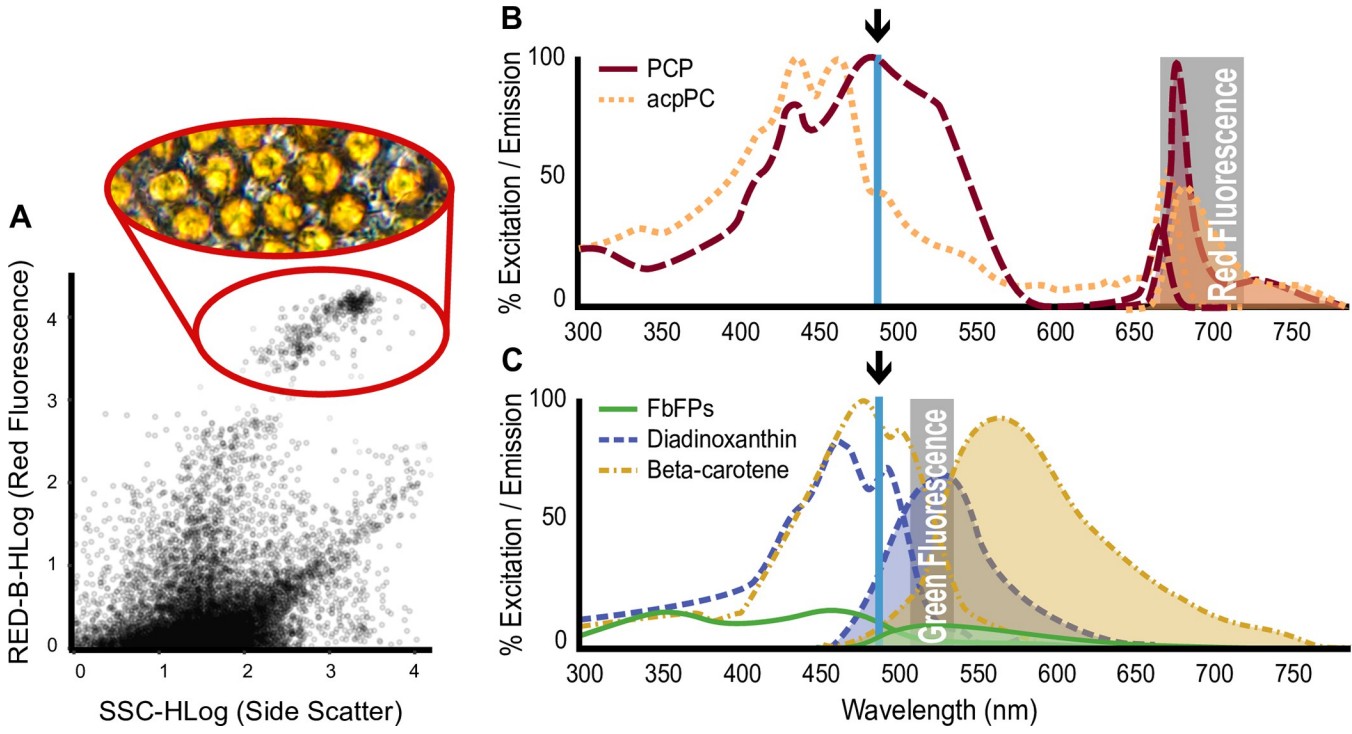

**Fig 1. A)** Sample point cloud visualizing all cytometry-detected particles within a well-replicate, and demonstrating how Symbiodiniaceae cells are identified based on high red fluorescence (**red circle**). **B, C)** Excitation/emission spectra employed by the flow cytometry protocol presented in this contribution. Excitation laser wavelengths are indicated by arrows. Lines depict absorption spectra, while lines with shaded areas underneath illustrate the intensity of emission wavelengths based on the corresponding excitation laser. **B)** The blue laser (488 nm) simultaneously excites the two dominant light harvesting complexes (LHC) of Symbiodiniaceae (PCP and acpPC) and a set of antioxidant associated pigments: FbFPs, diadinoxanthin, and beta-carotene (**C**). The two groups fluoresce differently upon excitation, which are differentiated with two emission filters that detect red (695/50 nm) (**B**) and green (525/30 nm) (**C**) spectra (**grey boxes**).

(Luminex Corporation, Austin, TX) benchtop flow cytometer and is publicly available through protocols.io (dx.doi.org/10.17504/protocols.io.dm6gpjr2jgzp/v3).

## 2.2 Identifying signals of autofluorescence

Our flow cytometry protocol relies on the detection of autofluorescence through the excitation of Symbiodiniaceae cells with a blue (488 nm) laser and emission detectors in the green (525/30 nm) and red (695/50 nm) spectra, termed green fluorescence and red fluorescence, respectively (Fig 1B and 1C).

In other flow cytometry methods, researchers typically compensate for spillover and spectral overlap by using labeled antibodies as controls to compensate for cellular autofluorescence and other contaminating fluorophores. This makes sense when using flow cytometry to detect labeled cellular structures (e.g. induction of labeled proteins through heat shock), but the quantification of autofluorescence requires different techniques given the high protein diversity and spectral overlap within an excitation-emission profile [23]. By identifying all possible targets within Symbiodiniaceae [7, 30–32] and modeling their excitation and emission profile based on previously published literature [33–38], we can infer sources of autofluorescence.

Excitable pigments previously identified from Symbiodiniaceae include beta-carotene, chlorophyll *a*, chlorophyll $c_2$, diadinoxanthin, diatoxanthin, and peridinin [7, 31]. We also identified flavin-based fluorescent proteins (FbFPs) as a probable source of autofluorescence [30, 39, 40]. To model our autofluorescence targets and identify any risk of spectral overlap,

spectral properties of the peridinin-chlorophyll *a* light-harvesting antennae protein complex (PCP) and riboflavin (FbFPs), compensated for blue (488 nm) laser excitation, were visualized using FluoroFinder Spectra Viewer (https://app.fluorofinder.com/ffsv/svs/). Beta-carotene [34], diadinoxanthin [37] and the chlorophyll *a*-chlorophyll $c_2$-peridinin protein complex (acpPC) [38] spectra were mapped and normalized to each other using the FluoroFinder spectra and absorbance spectra from Bricaud et al. [41].

## 2.3 Optimization assays

Photopigments are labile molecules prone to degradation in ambient conditions [42], as are Symbiodiniaceae cells when prepared improperly (see Sections 3.2 and 3.3). Two simple assays were designed and performed to provide a framework for optimizing the protocol. In both optimization assays, ideal conditions (temperature, dilution, time, etc.) were selected based on distributions with the highest consistency and tightest variances. This included FSC (forward scatter representing cell size) and SSC (side scatter representing cell shape), which are not only valuable phenotypic traits, but can also be used to detect potential cell clumping and cell lysing, respectively [43–46].

1. Assay one was used to determine the impacts of time, temperature, and light conditions on samples and the resulting data. A single staghorn coral (*Acropora pulchra*) fragment was airbrushed with filtered, sterile seawater to create a 30 mL (~150,000 cells/mL) tissue slurry. The slurry was homogenized by vortexing and needle shearing (S1 File), then equally distributed across four 50 mL falcon tubes. One mL of the tissue slurry was aliquoted from each falcon tube and immediately processed using our flow cytometry protocol (S1 File). After aliquots were removed, falcon tubes were separated and exposed to one of four conditions: (1) dark on ice, (2) dark at room temperature (22˚C), (3) ambient light on ice, and (4) ambient light at room temperature. A 1 mL sample from each falcon tube was processed approximately every two hours for a total of eight hours; a final set of samples was processed after being stored in their respective conditions overnight. This yielded six temporal samples processed 43, 136, 236, 344, 470, and 1459 minutes after removing tissue from the skeleton. At each time point, treatment replicates were loaded across three wells, and each well was sampled by the flow cytometer twice, creating six technical replicates per sample. Pairwise Dunn's tests (Z) were calculated using the FSA R package v0.9.3 [47], then organized into a compact letter display using the cldList command from the rcompanion package v2.4.21 [48].

2. Assay two was used to optimize tissue slurry dilutions and to identify possible effects of flow cytometer run times. We prepared six 1 mL (~150,000 cells/mL) tissue slurries from a single *Acropora pulchra* fragment for flow cytometry using our protocol (S1 File). Before loading samples into a 96-well microwell plate, all processed tissue slurries were combined into a single 50 mL falcon tube, needle sheared, and vortexed to homogenize samples to avoid possible batch effects. 50x, 20x, 10x, 5x, 2x, and 1x dilutions of the combined tissue slurry were prepared directly in the 200 uL wells. This created a total of 16 dilution series replicates, processed in sequential order until all 96 wells of the plate were filled. The flow cytometer processed each replicate dilution (2 within well technical replicates) at ~5 min intervals leading to breaks of 30–40 minutes between wells of the same dilution. until the run was complete. Time between replicates and dilutions are approximate as sample acquisition varies depending on the concentration of the sample.

## 2.4 Example application

In addition to the optimization assays, we performed an acclimation experiment with the upside-down jellyfish *Cassiopea*, a model organism often used to better understand bleaching and symbiosis [49], to demonstrate data handling techniques and discuss data interpretation. Clonal *Cassiopea* were placed into three different conditions for three weeks each. Clonal *Cassiopea* were obtained from UnderWater World Aquarium in Tumon, Guam. *Cassiopea* were transferred to flow through tanks at the University of Guam Marine Laboratory in Mangilao Guam on April 1, 2022 and allowed to acclimate until the start of the experiment on June 8, 2022. Jellyfish were 55–79 mm in diameter at the start of the experiment. Group 1 (n = 4) was placed in low-light conditions (50% shade), group 2 in high-light conditions (no shade) (n = 4), and group 3 experienced variable light conditions (n = 5) rotating from high-light to low-light to high-light (one week per condition). At the end of three weeks, digitate cirri (clusters of tentacle-like structures attached to the top of each oral arm) [50] were sampled with sterile scissors from each individual and placed in 1 mL of filtered seawater. Symbiodiniaceae cells were subsequently extracted and processed following the flow cytometry protocol described here (S1 File).

After identifying the Symbiodiniaceae cells based on red autofluorescence (Fig 1A), particulate noise was removed from the dataset (RED-B-HLog < 3.04001) yielding sample sizes of 16,134, 18,464, and 19,821 single-cell Symbiodiniaceae fluorescent profiles for dark, light, and variable treatment groups, respectively. To better understand data structure, power, and effect size, the effect of experimental conditions was analyzed at two resolutions: (1) phenotypic profiles of Symbiodiniaceae within each condition visualized and analyzed as entire assemblages, independent of host (n = 16,134, 18,464, 19,821) and (2) Symbiodiniaceae assemblages averaged to their respective host jellyfishes before visualizing and analyzing the effect of treatments (n = 4, 4, 5). A principal component analysis was performed and results plotted using the base stats package in R [51] to visualize the phenotypic variation associated with each experimental group. At both resolutions treatment groups were compared by using non-parametric Kruskal-Wallis tests ($X^2$) followed by post-hoc pairwise Dunn's tests (Z) using the FSA R package v0.9.3 [47]. All data processing and analysis for both of the optimization assays and the example application was completed with R v4.1.2 [51] in RStudio v1.3.1073 [52]. Figures were generated and modified with a combination of ggplot2 v3.3.5 [53] and InkScape v1.1 (https://inkscape.org).

All code and datasets required to replicate optimization assay and example application experiment analyses have been deposited on a public GitHub repository (https://doi.org/10.5281/zenodo.8260110).

## 3. Results

### 3.1 Fluorescent pigment identification

Red light emission caused by excitation with a blue laser is most likely caused by the core light harvesting complexes (LHC), peridinin-chlorophyll *a* light-harvesting antennae protein complex (PCP) or the chlorophyll a-chlorophyll *c2*-peridinin protein complex (acpPC) (Fig 1B), which use peridinin as the primary light harvesting molecule before efficiently passing it to chlorophylls [54]. This is in agreement with current literature that links red fluorescence off a 488 nm laser to chlorophyll [10, 11, 18, 19]. However, we suggest that due to chlorophylls intrinsic association with peridinin in Symbiodiniaceae [38, 55], we suggest identifying this fluorescence signature as LHCs or LHC-associated photopigments as it is more conservative and accurate for Symbiodiniaceae.

Increased green fluorescence in dinoflagellates under stress has been attributed to beta-carotene [11]. However, diadinoxanthin, diatoxanthin [37, 56], and flavin-based fluorescent proteins (FbFPs) [30, 39, 40] also emit green fluorescent light when excited by blue light (Fig 1C). We interpret green fluorescence as the combined signature of beta-carotene, xanthophylls (diadinoxanthin and diatoxanthin), and flavins given their fluorescent spectral overlap (Fig 1C).

## 3.2 Effects of sample preparation on degradation

Samples stored on ice and processed within approximately two hours (136 minutes) showed the most consistent signatures across all measurements (Fig 2). Red fluorescence degraded across all groups, but remained the most stable in the dark on ice, specifically for samples prepared within the first two time points (136 minutes) (Fig 2A). Green fluorescence (Fig 2B), forward scatter (Fig 2C), and side scatter (Fig 2D) were relatively stable compared to red

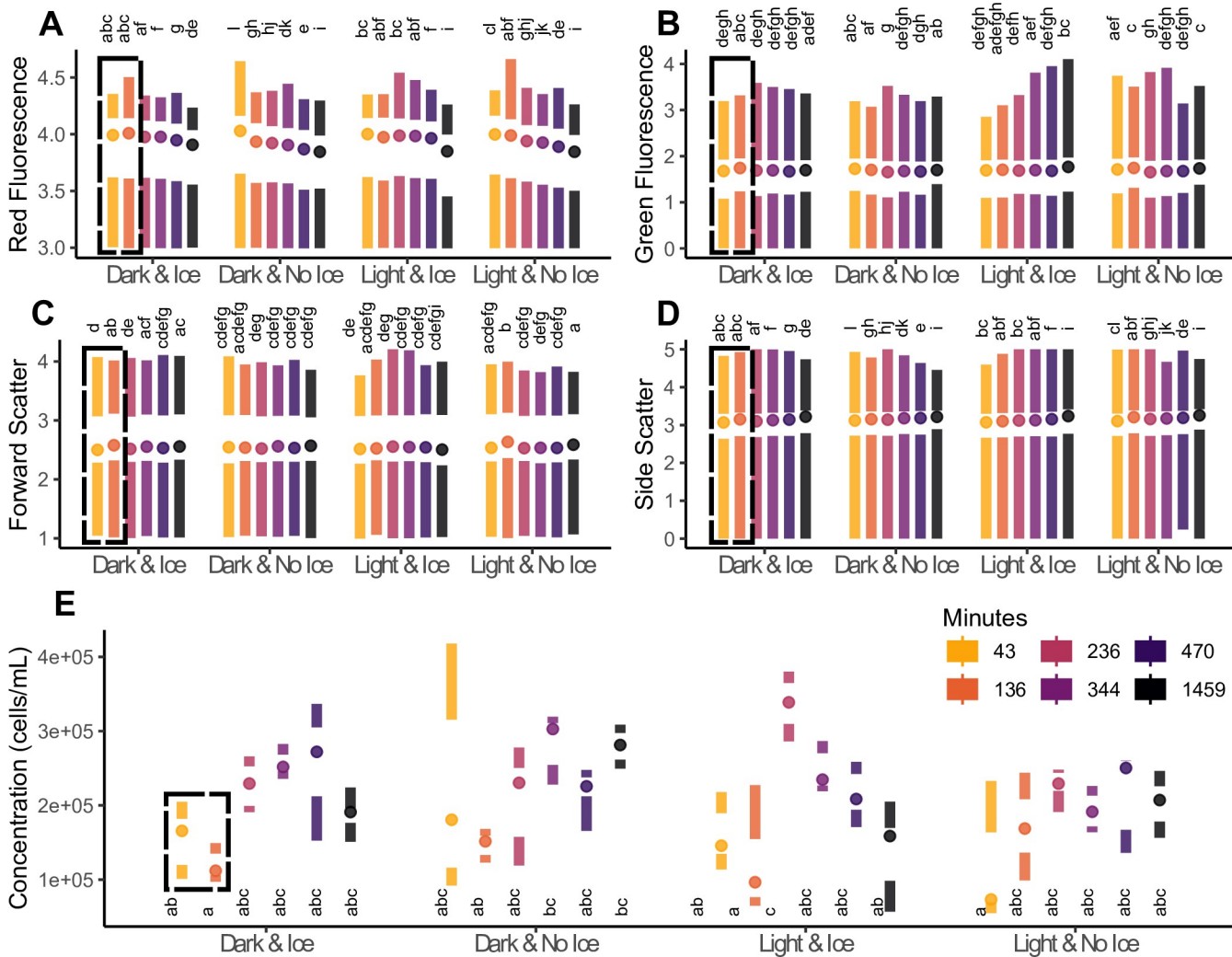

**Fig 2. Samples prepared under different light (Dark vs Light) and temperature (Ice vs No Ice) conditions.** Samples processed on ice within roughly two hours (136 minutes) of airbrushing yielded the most consistent results (**dashed boxes**), as informed by the statistical groups (p < 0.05) that are represented by letters **A**) Red fluorescence degraded quickly. **B**) Green fluorescence and forward scatter (cell size) (**C**) remained relatively stable over time. **D**) Side scatter (cell roughness), increased with time **E**) Cell concentrations were relatively consistent results when processed within 136 minutes.

fluorescence, although slightly higher readings in side scatter could indicate some cell lysing after prolonged preparation times (Fig 2D). Cell concentrations were consistent across all treatments when processed within 136 minutes (Fig 2E).

### 3.3 Effects of dilution and flow cytometer run-time on degradation

Six dilutions (50x, 20x, 10x, 5x, 2x, and 1x) were tested 16 times each across a 96-well plate resulting in individual sample measurements of ~150,000 cells/mL for each replicate (Fig 3). The flow cytometer run took around six hours to complete. Ten-fold and five-fold dilutions in the first four rows of the 96-well plate were the most consistent. More diluted samples produced unstable means of measured parameters, while less diluted samples exacerbated degradation. Red fluorescence decreased with time (Fig 3A), while green fluorescence increased with time (Fig 3B). Cell size (forward scatter) remained relatively consistent, but forward scatter increased in highly concentrated samples, overestimating cell size likely due to cell clumping (Fig 3C). Cell shape (side scatter) was the most stable metric suggesting low rates of cell lysis, though increasing quartile ranges in the one-fold and two-fold dilutions suggested increased susceptibility to lysis in higher concentrations (Fig 3D). Estimates of cell concentrations were heavily affected by dilution and run-time (Fig 3E).

### 3.4 Upside-down jellyfish light acclimation experiment

After three weeks, *Cassiopea* medusae visibly changed color with dark-acclimated individuals being dark brown and light-acclimated individuals being light brown. Alongside a visible change in medusa color, PCA indicated a substantial phenotypic shift within the Symbiodiniaceae assemblage, especially between the light and dark acclimated individuals, while the variable light exposure group showed a more mixed signal (Fig 4A). Treatment influenced fluorescence readings for all excitation/emission parameters, but statistical power varied depending on whether phenotypic variation was compared at the level of individual Symbiodiniaceae cells (Fig 4B–4E) or at the level of mean phenotypic variance within each host (Fig 4F–4I). Light-acclimated individuals had the lowest red fluorescence (LHC photopigments), forward scatter (cell size), and side scatter (cell shape), while dark-acclimated individuals had the lowest green fluorescence signal ('antioxidant' pigments) (Fig 4B–4I). When all Symbiodiniaceae measurements were included and compared across treatments (Fig 4B–4E), Dunn's tests identified statistically significant differences (p < 0.001) in most pairwise comparisons of treatments, despite small effect sizes. When averaging phenotypic variance to the jellyfish host individuals (n = 4 or 5), statistics were less powerful, but treatment effects became more conspicuous (Fig 4F–4I).

## 4. Discussion

### 4.1 Symbiodiniaceae fluorescence, light scatter, and phenotype

The purpose of the described methodology is to rapidly generate a phenotypic profile for thousands of individual Symbiodiniaceae cells, not necessarily to identify underlying mechanisms of phenotypic acclimation (e.g. photopigment regulation), although there is a substantial body of literature that uses flow cytometry to directly quantify chlorophyll or other pigments [10, 11, 17, 19, 21, 22, 57, 58]. In our example experiment (Fig 4), we revealed differences in Symbiodiniaceae phenotypes associated with different light conditions (Fig 4A). Symbiodiniaceae do not possess chlorophyll *b* [7, 31], a chlorophyll molecule common in other photosynthetic organisms that would normally be detected by our excitation-emission profiles. Instead, Symbiodinaceae possess two major LHC antennae, the peridinin-chlorophyll *a* protein complex

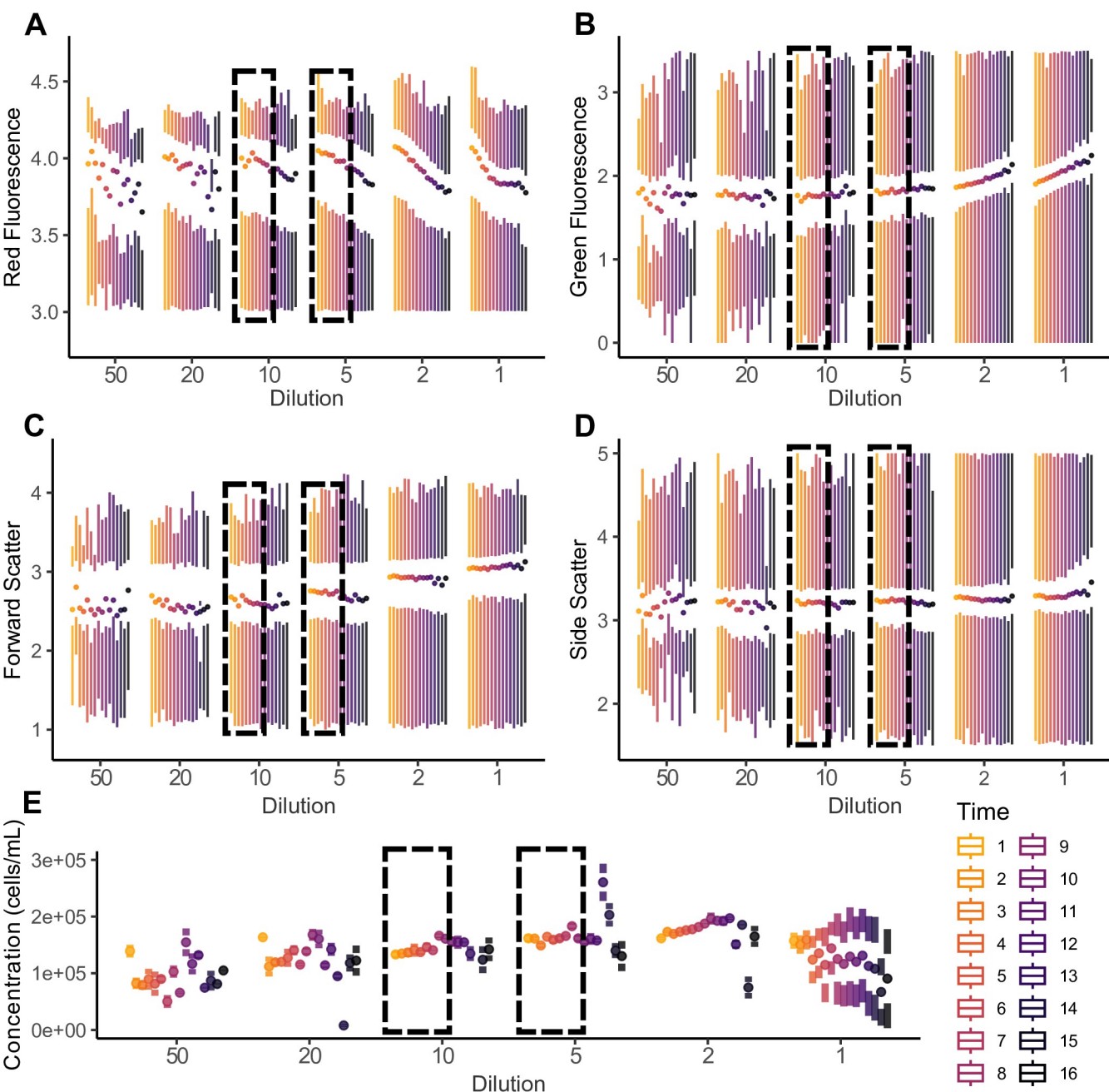

**Fig 3. Six dilutions (50x, 20x, 10x, 5x, 2x, and 1x) were tested 16 times each across a 96-well plate.** The cytometry run took ~6 hours estimating ~20 minutes between each of the 16 replicates (**Time**). Ten-fold and five-fold dilutions were the most consistent for half of the plate (**dashed boxes**) suggesting that runs should be limited to $\leq$ 48 wells (half of a standard 96-well plate). Over- or under-dilution of samples had a large effect on resulting parameter estimates. **A**) Red fluorescence degraded over time and had more variation with 50x and 20x dilutions. **B**) Green fluorescence increased over time, presumably due to heat generated by the flow cytometer. **C**) Cell size (FSC) did not change over time, but highly concentrated samples led to an overestimation of cell sizes, likely due to cell clumping. **D**) Cell shape (SSC) was the most stable parameter, suggesting low rates of cell lysis. **E**) Cell concentrations were heavily affected by dilution and time spent in the flow cytometer.

(PCP) and the chlorophyll *a*-chlorophyll $c_2$-peridinin protein complex (acpPC), which use peridinin and chl $c_2$ as the primary light-harvesting pigments [54, 59, 60]. Understanding Symbiodiniaceae-specific photosynthetic structures may influence flow cytometry data

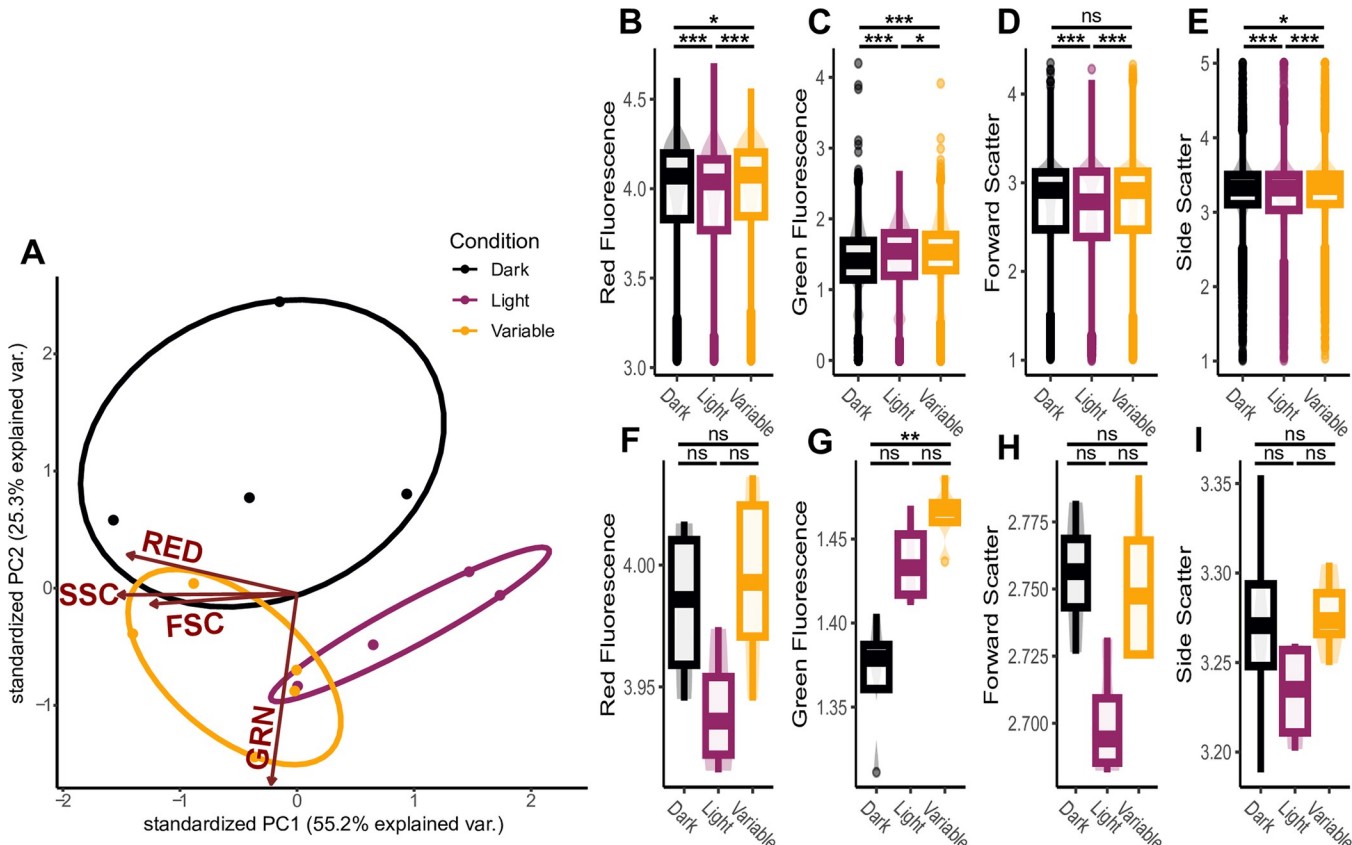

**Fig 4. Phenotypic profiles built from red fluorescence, green fluorescence, forward scatter, and side scatter for upside-down jellyfish processed with our flow cytometry protocol (S1 File) after three weeks of exposure to different light conditions (Dark, Light, and Variable). A)** Principal component analysis demonstrating divergence of phenotypic profiles associated with jellyfish of different experimental light conditions. **B-I)** Distributions for each phenotypic trait used to generate a phenotypic profile. Lines with asterisk indicate pairwise comparisons tested with Post-Hoc Dunn's tests ranging producing three categories for p-values: p > 0.05 (ns), p < 0.05 (*), and p < 0.001 (***). **B-E)** Each distribution represents 16,000–20,000 observations, representing the true phenotypic variance within a Symbiodiniaceae assemblage, which allowed for discriminating between treatments, even with small effect size. **F-I)** Phenotypic measurements averaged to the jellyfish host to better resolve conditional effect, despite decreased statistical power.

interpretation and inferences of phenotypic differences. For example, based on our modeled excitation/emission spectra, red fluorescence most likely reflects the relative quantity of LHC antennae or associated light harvesting pigments, primarily peridinin (Fig 1B). This is consistent with other studies that refer to this excitation/emission signature as chlorophyll [11, 17–19, 58], using flow cytometry for direct quantification of relative pigment abundance [10, 21, 22]. Our review of Symbiodiniaceae LHC composition and subsequent spectral mapping indicates that red fluorescence is likely targeting peridinin within peridinin-chlorophyll protein complexes (Fig 1A). While red fluorescence may indicate relative pigment abundance, it is possible that changes in red fluorescence are caused by changes in chloroplast size or location [61]. The tight association of peridinin with chlorophylls in LHC antennae [35, 54, 59, 60, 62, 63] and the extensive literature using flow cytometry to quantify chlorophyll or other pigments [10, 11, 17, 19, 21, 22, 57, 58] supports an interpretation of the reorganization of LHC protein complexes and their associated photopigments. For example, a conservative interpretation of red fluorescence in our *Cassiopea* acclimation experiment (Fig 4) is as follows: 'The reduction in the intensity of red fluorescence in high light could indicate the downregulation of photosynthetic pigments within Symbiodiniaceae cells to reduce light-associated stress'. This

interpretation is consistent with the down regulation of PCP that has been associated with light and heat stress [64].

Other studies that have measured green autofluorescence with flow cytometry identified it as beta-carotene [11, 23, 65]. However, after reviewing the available literature [7, 30, 34, 37, 38], we suggest that the intensity of green fluorescence represents a combination of photo-protective and antioxidant pigments and proteins that play primary roles in photo-acclimation and stress mitigation (Fig 1B). Beta-carotene is an efficient reactive oxygen species (ROS) scavenger, especially in the vicinity of high concentrations of ROS [66–68]. Diadinoxanthin and diatoxanthin are the main components in the photoprotective xanthophyll cycle, dissipating excess energy through non-photosynthetic quenching [56], alongside production of antioxidants [69]. The most abundant FbFPs (cryptochromes and riboflavin) act as key stress regulators [70, 71] and induce the accumulation of antioxidants [72–74]. Perhaps the relatively high intensity of green fluorescence in high light and variable light jellyfish individuals indicates the upregulation of antioxidant associated photopigments within Symbiodiniaceae cells to reduce light-associated stress (Fig 4).

The protocol presented here targets key LHC pigments (red fluorescence) and a combination of proteins known for their role in photo-acclimation and stress mitigation (green fluorescence) (Fig 1C) in conjunction with well-established measurements reflective of cell size (forward scatter) and cell shape (side scatter) [43, 44, 46, 75] to generate phenotypic profiles of Symbiodiniaceae. Independent of the underlying pigments, cell size, or cell shape, readings from the flow cytometer reflect phenotypes of individual cells, which may be influenced by the state of the cell (e.g. its progression through the reproductive cell cycle [29]). Large sample sizes, as generated by flow cytometry, enable the estimation of the average phenotypic profile within a Symbiodiniaceae assemblage. The protocol presented here represents a rapid method for high-throughput phenotypic profiling of Symbiodiniaceae assemblages, allowing for the estimation of phenotypic variance of Symbioodiniaceae communities and identification of average phenotypic difference across Symbiodiniaceae assemblages sampled from different hosts or experimental conditions (Fig 4), filling a gap in existing methodologies.

## 4.2 Protocol performance and optimization

Photopigments are labile molecules prone to degradation in ambient conditions [42], as is cell morphology when prepared improperly (Figs 2 and 3). Therefore, it is important to work consistently and expediently. This includes maintaining a consistent processing environment, sample concentrations and quantities, sample preservation methods, gain settings and calibration of the flow cytometer. Samples processed at room temperature were prone to rapid degradation, yielding imprecise estimates of fluorescent parameters compared to consistent autofluorescence-based phenotypic profiles for samples stored and processed on ice (Fig 2). However, autofluorescence signatures still changed after tissue removal from coral skeletons when processing was delayed, even when stored on ice. Based on our observations (Fig 2), we recommend that samples are processed for flow-cytometry within two hours of beginning sample preparation.

Cell concentration and flow-cytometer run-time are further factors to consider. Based on our assay (Fig 2), 5-fold to 10-fold dilutions of a sample with a starting concentration of ~150,000 cells/mL produced the most consistent dataset (15,000–30,00 cells/mL) (Fig 3). Over-dilution led to variable means and high variances of fluorescent parameters, while under-dilution exacerbated degradation and may have caused cell clumping, as indicated by increasing forward and side scatters (Fig 3). Independent of concentration, samples began to degrade after processing four rows (48 wells) of the 96-well plate (Fig 3). Therefore, we

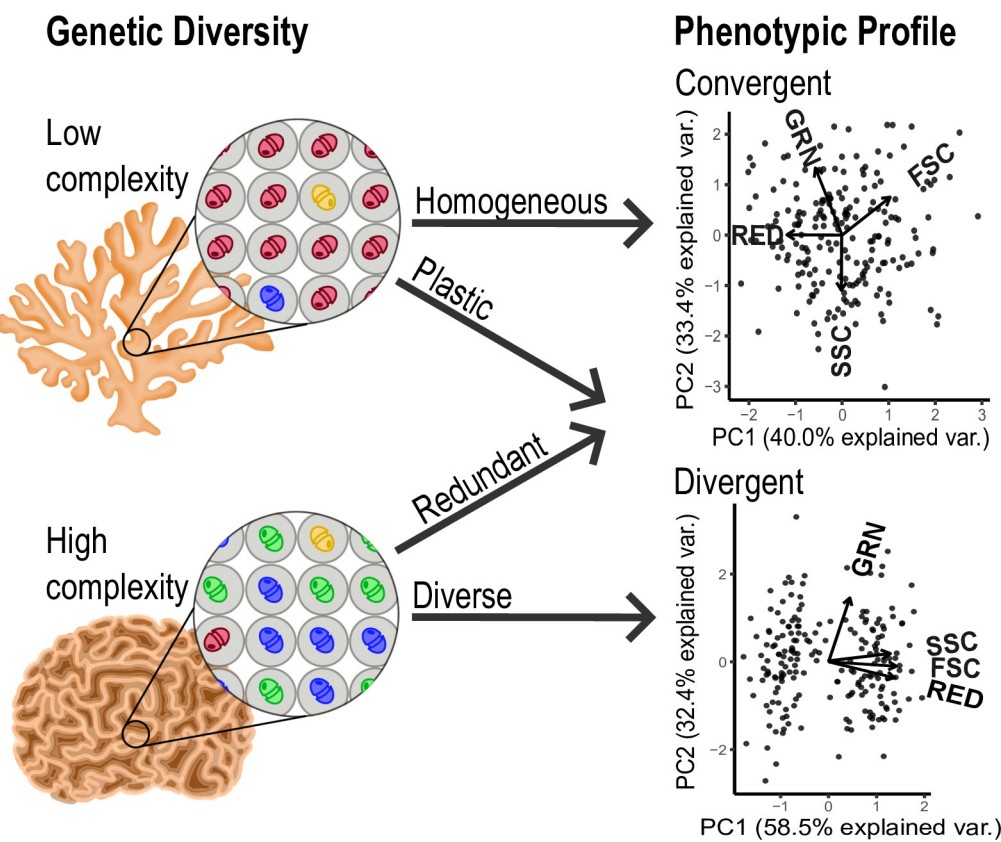

**Fig 5. Genetic diversity of Symbiodiniaceae assemblages *in-hospite* may reveal either low complexity (one dominant clade) or high complexity (more than one dominant clade) communities [3]; autofluorescence profiles may be either convergent or divergent, as shown here by PCAs consisting of the discussed phenotypic metrics.** As indicated by arrows, the combination of genetic diversity and phenotypic profiles has the potential to allow for the classification of Symbiodiniaceae assemblage functional strategies into homogenous, plastic, redundant, or diverse categories upon implementation into other relevant workflows. PCA datasets are theoretical and were generated and analyzed using R.

recommend loading no more than half of a 96-well plate per run with each well containing an estimated concentration of 15,000–30,000 cells/mL. None of the observed patterns were caused by settled cells, as each replicate was vortexed before being loaded into a microwell, and the flow cytometer automatically stirred each sample at a high velocity for seven seconds before data acquisition. The analyses presented here provide a roadmap for implementing and optimizing our flow-cytometry protocol. We recommend investing protocol optimization time upfront. Once established, our protocol should work across a diverse set of hosts, including corals, jellyfish, and hydroids, requiring little ongoing maintenance or cost.

Our *Cassiopea* acclimation experiment represents an example application of our protocol for comparisons of Symbiodiniaceae assemblages originating from different host individuals (Fig 4). The ability to detect extremely small, yet statistically significant effect sizes is a strength of flow cytometry, as it enables the rapid collection of large sample sizes. Large sample sizes increase statistical power, explaining why Symbiodiniaceae traits were statistically different across treatments despite seemingly small effect sizes when comparing measurements of Symbiodiniaceae cells directly between treatments (Fig 4A–4C). When averaging measurements from Symbiodiniaceae by host individuals, trends were more conspicuous but statistical power of our treatment comparisons weakened due to the low sample size of 4–5 host individuals per

treatment (Fig 4D–4F). Both levels of resolution are informative, but resolution of single Symbiodiniaceae cells is a more accurate representation of population-level phenotypic variance and provides greater statistical power for comparisons. We recommend researchers consider effect size and statistical power given their factorial resolution. The integration of the protocol presented here alongside other commonly used methodologies (e.g. PAM fluorometry, transcriptomics, or ITS2 metabarcoding) may help provide a more complete picture of Symbiodiniaceae acclimation and adaptation dynamics.

### 4.3 Applications

One difficult to answer question is whether changes in physiology of the Symbiodiniaceae assemblages are caused by phenotypic plasticity of individual Symbiodiniaceae cells or shuffling of Symbiodiniaceae assemblages to new functionally advantageous lineages. Integrating our flow cytometry protocol with Symbiodiniaceae metabarcoding would, for example, allow researchers to distinguish between adaptation through symbiont shuffling and acclimation through phenotypic plasticity (Fig 5). Symbiodiniaceae assemblages hosted in corals can either harbor homogeneous Symbiodiniaceae assemblages with low phylogenetic diversity (one dominant genus, species, or strain) or diverse assemblages with high phylogenetic diversity (several codominant genera, species, or strains) [3] (Fig 5). If a coral hosts a highly diverse Symbiodiniaceae assemblage, but the autofluorescence profile detected using our flow cytometry protocol is homogeneous, one may assume functional redundancy of symbiont clades or mediation of Symbiodiniaceae phenotypes by the coral host (Fig 5). Alternatively, if a coral hosts a low diversity Symbiodiniaceae assemblage, but multiple distinct phenotypic profiles are detected, one may assume phenotypic plasticity within the assemblage (Fig 5). Using our protocol for such applications has the potential for providing in-depth understanding of acclimation and adaptation dynamics of corals.

## Supporting information

**S1 File. Protocol for phenotypic profiling of endosymbiotic dinoflagellates using the Guava Flow Cytometer.** Also available on protocols.io: dx.doi.org/10.17504/protocols.io.dm6gpjr2jgzp/v3.
(PDF)

## Acknowledgments

We would like to thank Rebecca Salas for her assistance running the upside-down jellyfish acclimation experiment and MacKenzie Heagy for illustrating the coral-dinoflagellate assemblages shown in Fig 5.

## Author Contributions

**Conceptualization:** Colin Jeffrey Anthony, Colin Lock, Bastian Bentlage.

**Data curation:** Colin Jeffrey Anthony.

**Formal analysis:** Colin Jeffrey Anthony.

**Funding acquisition:** Bastian Bentlage.

**Investigation:** Colin Jeffrey Anthony, Colin Lock.

**Methodology:** Colin Jeffrey Anthony, Colin Lock.

**Project administration:** Bastian Bentlage.

**Resources:** Colin Lock, Bastian Bentlage.

**Software:** Colin Jeffrey Anthony.

**Supervision:** Colin Lock, Bastian Bentlage.

**Validation:** Colin Jeffrey Anthony, Colin Lock, Bastian Bentlage.

**Visualization:** Colin Jeffrey Anthony.

**Writing – original draft:** Colin Jeffrey Anthony.

**Writing – review & editing:** Colin Jeffrey Anthony, Colin Lock, Bastian Bentlage.

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
