## [Decision Letter · Decision Letter 0]

3 Mar 2023

PONE-D-22-33653High-throughput physiological profiling of endosymbiotic dinoflagellates (Symbiodiniaceae) using flow cytometryPLOS ONE

Dear Dr. Anthony,

Thank you for submitting your manuscript to PLOS ONE. After careful consideration, we feel that it has merit but does not fully meet PLOS ONE’s publication criteria as it currently stands. Therefore, we invite you to submit a revised version of the manuscript that addresses the points raised during the review process.

Thank you for your patience with your submission. One thing to really pay attention to with your resubmission is making sure that you have accurately cited the literature. For example, you are missing the Rosental et al, 2017 paper which first showed the utility of flow cytometery and non-stain assays on the coral and the symbiont cells. Additionally, reviewers brought up that you need to make sure that you do not "oversell" your work. While there are novel aspects, this is building on previous work, and it is important to acknowledge that. Also without other methods to confirm what you have found you must be careful with to not overstate your results. We look forward to seeing your resubmission. Please ensure that your decision is justified on PLOS ONE’s publication criteria and not, for example, on novelty or perceived impact.

We look forward to receiving your revised manuscript.

Kind regards,

Nikki Traylor-Knowles, Ph.D.

Academic Editor

PLOS ONE

Journal Requirements:

2."Please update your submission to use the PLOS LaTeX template. The template and more information on our requirements for LaTeX submissions can be found at " ext-link-type="uri" xlink:type="simple">http://journals.plos.org/plosone/s/latex."

"We would like to thank Rebecca Salas for running the upside-down jellyfish acclimation

experiment. We would also like to thank MacKenzie Heagy for illustrating the coral dinoflagellate assemblages shown in Fig 5. Guam NSF EPSCoR directly supported this work through the National Science Foundation award OIA-1946352. Any opinions, findings, conclusions, or recommendations expressed in this contribution are those of the authors and do

not necessarily reflect the views of the National Science Foundation"

"CJA, CL, and BB are all directly supported by NSF Guam EPSCoR (https://guamepscor.uog.edu/) through the National Science Foundation (https://www.nsf.gov/) award OIA-1946352. The funders had and will not have a role in study design, data collection and analysis, decision to publish, or preparation of the manuscript"

Reviewers' comments:

Reviewer's Responses to Questions

**Comments to the Author**

1. Does the manuscript report a protocol which is of utility to the research community and adds value to the published literature?

Reviewer #1: Yes

Reviewer #2: Yes

2. Has the protocol been described in sufficient detail?

To answer this question, please click the link to protocols.io in the Materials and Methods section of the manuscript (if a link has been provided) or consult the step-by-step protocol in the Supporting Information files.

The step-by-step protocol should contain sufficient detail for another researcher to be able to reproduce all experiments and analyses.

Reviewer #1: No

Reviewer #2: Yes

3. Does the protocol describe a validated method?

Reviewer #1: No

Reviewer #2: Yes

4. If the manuscript contains new data, have the authors made this data fully available?

Reviewer #1: Yes

Reviewer #2: Yes

**5. Is the article presented in an intelligible fashion and written in standard English?**

Reviewer #1: Yes

Reviewer #2: Yes

6. Review Comments to the Author

Reviewer #1: The authors present some groundwork in developing new protocol points and a new application of flow cytometry in understanding the photosynthetic response of Symbiodiniaceae. As someone who has worked on developing new technologies and uses for flow cytometry I found the premise very interesting. However, ultimately he paper fell short, and/or the authors over-reached, in presenting the data generated. My main issues concern the following (1) for a new method which makes big statements about its use in photo physiology, there is no ground truthing (i.e., comparative HPLC or even simple chl extractions), (2) this is extra problematic for flow cytometry where there is inability to translate RFUs to biologically meaningful units or understand effect sizes, (3) there is almost no acknowledgement of the potential issues with this protocol/interpretation – e.g., fluorescence spill over, the sensitivity of flow cytometers to gain settings. In the face of these issues, the very specific and detailed statements that the authors make to relate them to photophysiology are not well enough supported. I think the paper would need major revisions to reflect what are interesting hypotheses vs. proof of concept.

In a lesser issue, the paper has a mixed identity of two very distinct projects that are never tied together well: a methodological step in preservation and cell counting, and a deeper dive into autofluorescence and functional profiling.

The first aspect, methodology is relatively straight forward. I appreciate having some parameters for best practices with pigment analysis and cells counting. One thing the authors should consider though is that cells settle during processing, and while waiting to be processed. So best practices are normally to vortex the sample right before loading into the flow cytometer so that the sample is homogenized and the counts are accurate for doing a concentration by run time * flow rate. Doing the whole sample is also fine but you run into issues, as did the authors, with the clumping of settled cells and variable counts for samples coming later in the processing. Please consider this in the discussion.

The second aspect, empirical application, has several issues, that unfortunately make it unsuitable for publication. Some of these were stated above. Firstly, from my experience, there is quite a lot of spectral overlap of pigments across channels, but no clear discussion of this or how compensation was done.

The analysis of the data with respect to the jelly experiment is unclear. There is no indication of how the 4 samples were dealt with in the analysis. Is Fig. 4 one sample from one jelly in each treatment? Samples from multiple jellies plotted together? How was the gain accounted for? Did the analysis take into account that the experimental replication was 4 not 20,000? The differences between treatments seem so small even on a log scale against RFUs and the extremely low p-values, despite little visible grouping suggests that this is being driven by cell numbers rather than experimental treatments. If you randomly sampled 100 cells, is there still a difference?

Can the authors include some of the plots used for gating forward and sidescatter, particularly those used to “remove noise”. The results described in lines 196-203 are difficult to parse. Statistics could be included where the trends are mentioned rather than before.

In the fig 4 caption, the authors acknowledge that the effect size is small but the following discussion misses this point entirely.

The discussion is just largely overinterpreted with no acknowledgements of the major issues I mentioned above. Even though I enjoyed reading it in parts and thinking about these hypotheses in terms of shifts in pigments that might happen, the data are just not there to support it given the lack of groundtruthing, the small effect size, and the inability to make biological sense of RFU’s.

The end of the discussion, line 281. This paragraph is missing some context. There are papers that use flow cytometry to identify genetic groups and look at nutrients (McIlroy et al. 2020), it’s missing some cool flow cytometry work on functional groups across depth from Apprill et al. 2007 Coral Reefs; there is some great cell cycle work recently in tivey et al. 2020. There is more to functional diversity then pigment content.

Fig. 5 - Are these PCA’s real or theoretical? If theoretical why not plot the high and low light groups generated from this. Please be explicit about this.

Reviewer #2: In the manuscript entitled: "High-throughput physiological profiling of endosymbiotic

dinoflagellates (Symbiodiniaceae) using flow cytometry" Anthony et al, describe a new way of flow cytometry analysis of the symbiont algae in corals. What is unique in this work is while flow cytometry was used in the past for the algae analysis, they take the fluorescent properties to analyze the state on a single cell level, which flow cytometer is enabling. Interestingly they are doing it without adding too much additional tools, making this method ready to be applied in many labs around the world.

This is a well written manuscript with a good and applicable idea, which can be used in many laboratories working on algae in different organisms and not only corals.

This reviewer has only few minor comments:

Figure 5 - Labels of 'Low complexity' and 'High complexity' are both too close to the branching coral diagram and should moved to avoid confusion. I would suggest moving each label underneath its relevant diagram.

Methods - Line 82 - comma needed after "laser excitation".

Methods - Line 102 - it is not clear how the authors prepared the plate.

" three across-well and two within-well replicates were used for data collection"

What does that mean? How many wells were used and how many replicates?

General Question: Please mention how you kept the experimental animals? Like a 'Animal Husbandry' section in the methods?

Methods - Line 103 - 111 (Assay 2) - Initial cell concentration should be noted in the methods.

(It appears as a note in the supplementary protocol File S1)

(It also appears on line 168)

General Question about Figure 2: In the text the authors highlighted that certain conditions bring the most consistent results which were indicated by the dashed boxes. Please indicate how conclusions are determined.

7. PLOS authors have the option to publish the peer review history of their article (what does this mean?). If published, this will include your full peer review and any attached files.

Reviewer #1: No

Reviewer #2: No

---

## [Author Response · Author response to Decision Letter 0]

23 Jun 2023

A formatted letter (.pdf) has been included with the revised submission. Please see that document for specific and complete responses to the editor and reviewers. We look forwarded to seeing your response to this revised version.

---

## [Decision Letter · Decision Letter 1]

13 Aug 2023

Rapid, high-throughput phenotypic profiling of endosymbiotic dinoflagellates (Symbiodiniaceae) using benchtop flow cytometry

PONE-D-22-33653R1

Dear Dr. Anthony,

We’re pleased to inform you that your manuscript has been judged scientifically suitable for publication and will be formally accepted for publication once it meets all outstanding technical requirements.

Kind regards,

Nikki Traylor-Knowles, Ph.D.

Academic Editor

PLOS ONE

Additional Editor Comments (optional):

Reviewers' comments:

Reviewer's Responses to Questions

**Comments to the Author**

1. Does the manuscript report a protocol which is of utility to the research community and adds value to the published literature?

Reviewer #1: Yes

Reviewer #2: Yes

2. Has the protocol been described in sufficient detail?

To answer this question, please click the link to protocols.io in the Materials and Methods section of the manuscript (if a link has been provided) or consult the step-by-step protocol in the Supporting Information files.

The step-by-step protocol should contain sufficient detail for another researcher to be able to reproduce all experiments and analyses.

Reviewer #1: Yes

Reviewer #2: Yes

3. Does the protocol describe a validated method?

Reviewer #1: Yes

Reviewer #2: Yes

4. If the manuscript contains new data, have the authors made this data fully available?

Reviewer #1: Yes

Reviewer #2: Yes

**5. Is the article presented in an intelligible fashion and written in standard English?**

Reviewer #1: Yes

Reviewer #2: Yes

6. Review Comments to the Author

Reviewer #1: The authors worked hard and carefully to transform this manuscript for the better. The new context and framing is so much more clear. I found it interesting and I even learned something from their more thorough review of the literature. Rather than over-reaching the kept everything in context. I think people will find the paper overall very useful. The previous review I provided was probably not as positive as they would have hoped but I appreciate the authors responding to it. I really think the paper is much improved. I only have a minor comment about one sentence which is really just a suggestion but I believe the paper as is will make a great contribution.

In Line 37 - I'm being pedantic but you can also use PAM for cultured Symbiodiniaceae.

Reviewer #2: There are no further comments. The authors responded and amended all previous comments in full. The manuscript is approved by this reviewer.

7. PLOS authors have the option to publish the peer review history of their article (what does this mean?). If published, this will include your full peer review and any attached files.

Reviewer #1: No

Reviewer #2: **Yes: **Benyamin Rosental

---

## [Editor Report · Acceptance letter]

7 Sep 2023

PONE-D-22-33653R1 

Rapid, high-throughput phenotypic profiling of endosymbiotic dinoflagellates (Symbiodiniaceae) using benchtop flow cytometry 

Dear Dr. Anthony:

I'm pleased to inform you that your manuscript has been deemed suitable for publication in PLOS ONE. Congratulations! Your manuscript is now with our production department. 

Kind regards, 

on behalf of

Dr. Nikki Traylor-Knowles 

%CORR_ED_EDITOR_ROLE%

PLOS ONE